# Factors associated with change in adherence to COVID-19 personal protection measures in the Metropolitan Region, Chile

**Simón Varas**[1☉]**, Felipe Elorrieta**[1☉]*****, Claudio Vargas**[1☉]**, Pablo Villalobos Dintrans**[2‡]**, Claudio Castillo**[2‡]**, Yerko Martinez**[2‡]**, Andrés Ayala**[1‡]**, Matilde Maddaleno**[2‡]

**1** Departamento de Matemáticas y Ciencia de la Computación, Universidad de Santiago, Estación Central, Santiago, Chile, **2** Programa Centro Salud Pública, Facultad de Ciencias Médicas, Universidad de Santiago, Estación Central, Santiago, Chile

☉ These authors contributed equally to this work.
‡ PVD, CC, YM, AA, and MM also contributed equally to this work.
* felipe.elorrieta@usach.cl

**Data Availability Statement:** All relevant data are within the paper and its Supporting Information files.

## Abstract

Personal protective measures such use of face masks, hand washing and physical distancing have proven to be effective in controlling the spread of the Covid-19 pandemic. However, adherence to these measures may have been relaxed over time. The objective of this work is to assess the change in adherence to these measures and to find factors that explain the change For this purpose, we conducted a survey in the Metropolitan Region of Chile in which we asked the adherence to these measures in August-September 2021 and retrospectively for 2020. With the answers obtained we fit a logistic regression model in which the response variable is the relaxation of each of the self-care preventive actions. The explanatory variables used are socio-demographic characteristics such as the age, sex, income, and vaccination status of the respondents. The results obtained show that there has been a significant decrease in adherence to the three personal protection measures in the Metropolitan Region of Chile. In addition, it was observed that younger people are more likely to relax these measures. The results show the importance of generating new incentives for maintaining adherence to personal protection measures.

## 1. Introduction

Since 2020 the world has been affected by a pandemic caused by the SARS-CoV-2 virus that causes the Covid-19 disease. One of the first defenses against the transmission of this virus are the self-care preventive actions, such as the use of facemasks, hand washing, and keeping a physical distance, all recommended by the World Health Organization (WHO) and other health public institutions [1, 2]. These measures have proven to be very efficient in controlling the advance of this virus [3–5]. However, due to pandemic fatigue and the advent of vaccines, some evidence suggests that adherence to these measures may have relaxed during the course of the pandemic [6–8]. Adherence to these recommendations continues to be a challenge due to local inequalities, cultural differences and associated changes in people's behavior [9–14].

**Funding:** The authors received no specific funding for this work.

**Competing interests:** The authors have declared that no competing interests exist.

This study assesses adherence to self-care preventive actions in the Metropolitan Region of Chile. For this purpose, a survey was performed in which we asked for the adherence to the use of facemasks, hand washing, and keeping a physical distance in August-September 2021 and retrospectively during 2020.

This survey was implemented in a period of transition between two Covid-19 waves, according to the wave definition given by [15]. The weekly moving average of the confirmed cases of Covid-19 in the Metropolitan Region of Chile at that time was 4 cases per 100.000 inhabitants [16]. According to the guidelines of the Health Ministry of Chile, the Metropolitan Region had opened the lockdowns due to Covid-19.

Regarding vaccines, Chile has had a successful campaign [17], reaching more than 80% of the Chilean population fully vaccinated and more than 60% with a booster dose by December 2021. However, these vaccination levels have not been distributed homogeneously in the country. Particularly in the Metropolitan Region of Chile, a high level of heterogeneity between its municipalities has been observed both in the vaccination progress and in the number of people who have been infected.

Despite the progress of the vaccination campaign in Chile, greater protection can be achieved if the vaccine is complemented with self-care preventive actions [18]. Moreover, it has become very relevant to reinforce our protection due to the arrival of more contagious variants of SARS-CoV-2 such as delta or omicron.

In this context, it is interesting to assess whether the self-care preventive actions have decreased their adherence in Chile. Consequently, the main objective of this article is to assess the change in adherence to the COVID-19 personal protective measures in the Metropolitan Region and to identify factors associated that could explain the relaxation of these measures.

## 2. Methods and materials

### Study design and setting

This is a cross-sectional study, although with a longitudinal focus, since adherence to preventive measures was consulted at two different points in time. The period covered is 2020 and 2021 (August-September).The study was conducted in the Metropolitan Region of Chile, considering responses from 48 municipalities of the 6 provinces of this region.

This region was selected because it is the main region of Chile, where more than 40% of the total population of the country lives. In addition, in the first pandemic wave this region reached the highest mortality rate of the country.

In order to incorporate a spatial component to the analysis, the nearest health service according to the reported residence of the participant was considered. The Chilean health system uses a geographic administrative division to organize health services for the population. Thus, the Metropolitan Region has six major health service areas: north, west, center, east, south and south-east.

### Sampling method and sample size

A non-probabilistic purposive sampling was applied throughout the Metropolitan Region, considering responses from all provinces of the region. The sample size included 635 responses, 66 of which were validated by phone call, verifying the authenticity of the information provided. The sampling size was intentionally completed to meet quotas for geographic, socioeconomic and sex representation

### Data collection

The information was obtained through an online questionnaire, disseminated in the social networks between August 6 and September 8, 2021.

The survey inquired about the frequency of the application of three COVID-19 personal protection measures: use of facemasks, hand washing, and physical distancing. Participants were asked about the adherence at the current time (August-September 2021) and retrospectively for 2020. The frequency of implementation of preventive measures included: almost never, sometimes, most of the time, and always. In addition, the participants were asked about their age, sex, level of household income, area of residence and whether they had been vaccinated, they intended to do it, or did not want to get the vaccine.

To record the change between the years 2020 and 2021, a variable was created with three categories to identify individuals that increased, maintained, or decreased adherence to each of the COVID-19 preventive measures. The Wilcoxon signed-rank test for paired data [19] was used to assess whether the change in adherence in each measure was statistically significant. If the null hypothesis of this test is rejected then the reduction of adherence is significant.

### Data analysis

In order to explain the change in adherence to the COVID-19 preventive measures, we performed a logistic regression model for each self-care measure. For this purpose, the change in the adherence was dichotomized for each self-care measure taking the value 1 if a person increases or maintains adherence and as 0 if a person decreases its adherence. In order to assess the sensitivity of this categorization, we compared the results obtained with an ordinal logistic regression where the response variable was defined in the following order (increases adherence, maintains adherence, and decreases adherence).

For each logistic regression, a set of socio-demographic characteristics of the participants of the survey was included. The explanatory variables included the per capita household income (divided into three continuous groups; High socioeconomic group with a monthly income above 1,500,000 Chilean pesos, Medium with income between 1,500,000 and 500,000 Chilean pesos, and Low with income below 500,000 Chilean pesos), sex, age, vaccination (asked about having received at least one dose) and the nearest health service according to the reported residence of the participant.

The effect of the explanatory variables are presented in terms of Odds Ratio, and represents the probability of decreasing adherence versus the probability of maintaining or increasing adherence, where values close to 1 express equal probability or absence of effect, values above 1 indicate a positive effect, and coefficients below 1, a negative effect. Likewise, the confidence interval (CI) of each estimate is included with 95% statistical confidence ($p < 0.05$).

### Ethical considerations

The survey had the informed consent of the participants. Participation was completely voluntary, with no risks or benefits for the respondents. The information collected was handled anonymously and confidentially. The research was approved by the Ethical Review Committee of the Universidad de Santiago, Chile.

## 3. Results

The sample obtained in the survey is composed as follows: 56.8% reported themselves as women and 41.5% as men, while 1.7% reported themselves as neither of the above. The age range was between 18 and 74 years. Regarding monthly household income, 33% reported earning less than 500,000 Chilean pesos, 44% reported earning between 500,000 and 1,500,000 Chilean pesos, and 23% reported earning more than 1,500,000 Chilean pesos. When asked about vaccination status, 95.8% reported having at least one dose, while 3.8% reported not being vaccinated (either because they do not want to, because they have not yet decided to do

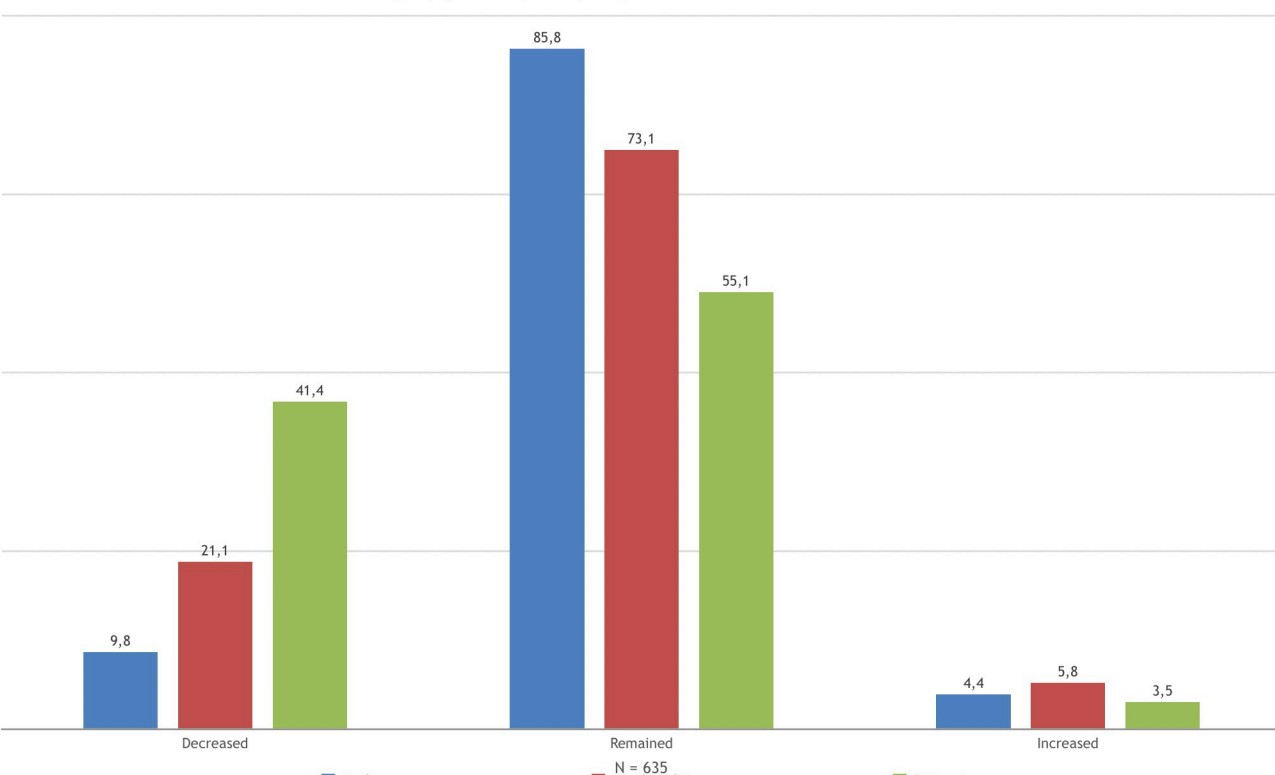

**Fig 1. Change (%) in the frequency of COVID-19 personal protection measures between 2020 and 2021.**

so, or because they are waiting to do so) and 0.5% marked don't know/no answer. In relation to the nearest metropolitan health service reported, 16.7% corresponded to the East, 8.5% to the North, 20.5% to the West, 19.2% to the Center, 13.4% to the South and 21.7% to the South-East.

In relation to the use of facemasks, Fig 1 shows that 4.4% of the sample increased their adherence between the years 2020 and 2021, 85.8% maintained it, and 9.8% decreased it. With respect to hand washing, 5.8% increased their adherence, 73.1% maintained it, and 21.1% decreased it. Finally, regarding physical distancing, 3.5% increased their adherence to this behavior, 55.1% maintained it and 41.4% decreased it. In other words, physical distancing is the measure with the highest decrease in adherence in the year 2021.

The p-values obtained from the Wilcoxon signed rank test for the three preventive measures were less than 0.001. Therefore, the reduction in adherence observed in the descriptive analysis is statistically significant.

Table 1 shows the logistic regression models fitted to explain the relaxation of the three COVID-19 personal protection measures. As can be seen, for the use of facemasks and hand washing, the household income, sex, vaccination, and the metropolitan health service had no significant effect. On the other hand, it was observed that the higher the age, the chances of decreasing the frequency of facemask use and handwashing decreased statistically significantly.

This age effect is also observed for physical distancing. However, we also found significant effects of sex and vaccination status in the relaxation of this measure. According to the odds ratio estimated, men have approximately a half probability of relaxing physical distancing than women. In addition, respondents that are unvaccinated have a lower probability to relax this

**Table 1. Logistic regression models for the change in the COVID-19 personal protection measures (facemask use; hand washing; distancing) between 2020 and august-september 2021.**

| Predictors | Facemask use | | Hand washing | | Distancing | |
|---|---|---|---|---|---|---|
| | Odds Ratio | CI | Odds Ratio | CI | Odds Ratio | CI |
| (Intercept) | 0.199 * | 0.052 – 0.712 | 1.250 | 0.488 – 3.206 | 2.860 * | 1.281 – 6.451 |
| Household income | 0.997 | 0.664 – 1.492 | 1.152 | 0.857 – 1.551 | 1.032 | 0.806 – 1.323 |
| Sex (Ref = Woman): Man | 1.449 | 0.833 – 2.513 | 0.726 | 0.473 – 1.104 | 0.531 *** | 0.374 – 0.751 |
| Age | 0.973 * | 0.948 – 0.997 | 0.960*** | 0.941 – 0.979 | 0.970 *** | 0.955 – 0.985 |
| Vaccine (Ref = Vaccinated): Unvaccinated | 2.005 | 0.562 – 5.634 | 0.899 | 0.254 – 2.495 | 0.322 * | 0.091 – 0.892 |
| MHS (Ref = East): North | 1.259 | 0.393 – 3.787 | 0.573 | 0.237 – 1.302 | 1.059 | 0.520 – 2.150 |
| MHS (Ref = East): West | 1.378 | 0.556 – 3.593 | 0.576 | 0.290 – 1.135 | 0.721 | 0.400 – 1.297 |
| MHS (Ref = East): Central | 1.044 | 0.413 – 2.719 | 0.540 | 0.277 – 1.043 | 0.806 | 0.459 – 1.415 |
| MHS (Ref = East): South | 0.975 | 0.339 – 2.760 | 0.542 | 0.256 – 1.118 | 0.798 | 0.426 – 1.487 |
| MHS (Ref = East): South-East | 0.808 | 0.304 – 2.166 | 0.736 | 0.393 – 1.378 | 0.835 | 0.480 – 1.451 |
| Observations | 622 | | 622 | | 619 | |
| $R^2$ Tjur | 0.015 | | 0.043 | | 0.071 | |
| Deviance | 389.753 | | 612.711 | | 794.436 | |
| AIC | 409.753 | | 632.711 | | 814.436 | |
| log-Likelihood | -194.876 | | -306.355 | | -397.218 | |

Note:

* $p<0.05$

** $p<0.01$

*** $p<0.001$

MHS: Metropolitan Health Service.

measure than the respondents that have at least one dose of the vaccine. Finally, the income and the nearest metropolitan health service show no significant effects.

The results presented above are consistent with those obtained in the ordinal logistic regression used to evaluate its sensitivity. For instance, in the ordinal logistic regression for both hand washing and physical distancing, it was found that younger people are more likely to relax adherence to preventive self-care measures. In addition, in the ordinal logistic regression for physical distancing we also found significant effects of the sex and the vaccination status. The only difference found in the sensitivity analysis was observed in the face mask use model, since in this model no significant variables were obtained for the ordinal logistic regression.

A more intuitive way to show these results comes from the estimated marginal probabilities for each model. We focus on calculating these probabilities for different ages since the effect of this variable was significant in all models. In Fig 2 we show these probabilities for the models of facemask use and hand washing. Note that in both cases as age increases, the probability of relaxing these measures decreases. More specifically, the probability of relaxing the mask use estimated by the logistic regression decreases from 0.13 in 18-year-olds to 0.04 in 74-year-olds, while for hand washing this probability decreases from 0.31 to 0.05.

For the physical distancing model, we calculated the marginal probabilities using age and sex since they showed a significant effect in the logistic regressions (Table 1). As presented in Fig 3, the predicted probabilities of relaxing physical distancing are consistently higher for women than for men. This relaxation can be understood by the fact that in the baseline (year 2020) women had a higher adherence to the physical distance than men, but by the year 2021

**Estimated probabilities of Age and Sex in the Physical distancing model**

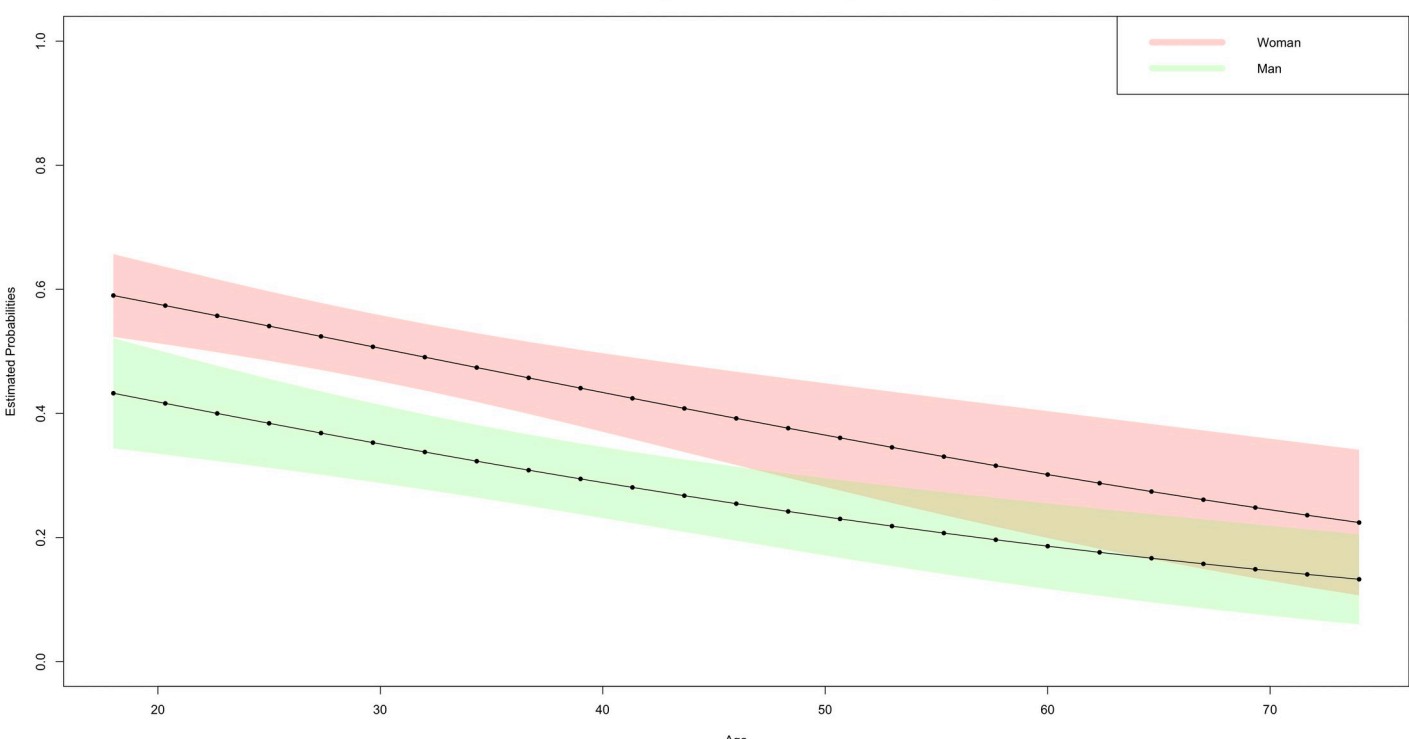

**Fig 2.** Probabilities estimated by logistic regression for age in the mask use (left) and hand washing (right) models with confidence intervals of 95%. The probabilities shown in the figure were estimated by a logistic regression fitted with only the significant variables at level 5%.

this adherence tends to equalize between both sexes. Moreover, for people between 20 and 45 years of age, the differences in the predicted probabilities for women and men are statistically significant. For women, the probability of relaxing physical distancing decreases from 0.6 at age 18 to 0.22 at age 74. On the other hand, the probability of relaxing physical distance decreases from 0.43 for 18-year-old men to 0.13 for 74-year-old men.

## 4. Discussion

This study aims to determine the relaxation of adherence to three preventive self-care measures for Covid-19 in the Metropolitan Region, Chile in August-September 2021 with respect to 2020, as well as the factors that explain this relaxation. The results obtained show that the three measures studied, use of facemasks, hand washing, and keeping a physical distance had lower adherence in 2021 than in 2020.

Regarding to the objective of assessing the factors associated with changes in adherence to personal protective measures, this study shows that the age of the respondents has a significant effect on this relaxation for the three measures studied so that young people are more likely to relax these measures, while older people were more likely to maintain adherence to preventive measures over time. This age effect has also been reported in other studies [6, 7, 20–24]. This effect may be explained by differences in the perception of the severity of the virus. Some studies point out that older people experience greater nervousness, perceive greater severity and take fewer risks in the face of the virus than younger people [25–27], although contradictory results have also been reported [28, 29].

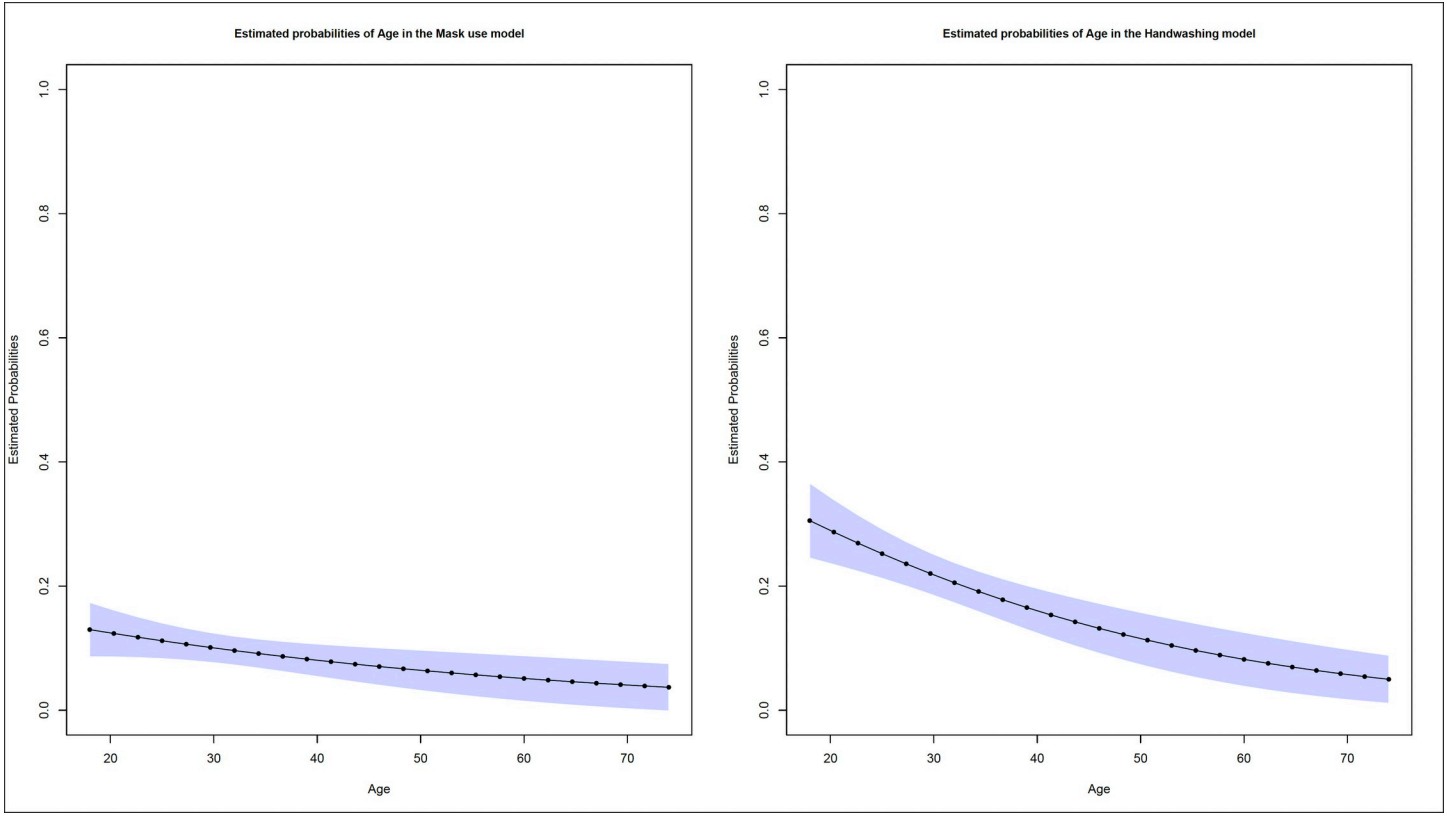

**Fig 3. Probabilities estimated by logistic regression for age and sex in the distancing model and with confidence intervals of 95%.** The probabilities shown in the figure were estimated by a logistic regression fitted with only the significant variables at level 5%.

Furthermore, for the physical distancing model, we also reported a statistically significant effect of the sex of the respondents. Particularly, a greater relaxation of physical distance was found in women. This is partly explained by the fact that women had a higher baseline adherence than men. Some studies have also reported the higher adherence to physical distancing of women in the pandemic first year [24, 30, 31]. However, the higher initial adherence of women led to a more pronounced relaxation until reaching a similar adherence to this measure in 2021 than men. Finally, people who have not been vaccinated are less likely to relax the physical distance. Household income and the nearest metropolitan health service did not show a significant effect on the response variable.

Some limitations of this study are the bounded information from the survey; we preferred to create an instrument with few questions to ensure a good response rate. A panel study could help monitoring and assessing these effects in the future. In addition, the proportion of vaccinated people in the sample was higher than that observed in the population. A challenge for further extensions of this work is to achieve more responses from unvaccinated persons, particularly since this showed to be a relevant factor explaining protective behaviors. In addition, as the survey was conducted in a period between two waves of covid, the participants' answers of this survey could be affected by a less risk perception of the people.

Finally, the application of an online survey introduces certain biases. Although it is estimated that Internet coverage has grown substantially in recent years, only 67.48% of the country's households have a fixed Internet connection [32]. Although this average percentage is higher in the Metropolitan Region, the Internet coverage gap between vulnerable households

and the most advantaged households is still very wide [33]. Consequently, there is a possibility that the results only represent only those who have access to the Internet. This issue was addressed by incorporating individuals from households with different socioeconomic incomes to meet quotas of all socioeconomic groups.

## 5. Conclusions

As long as there is no global control of the Covid-19 pandemic, the relaxation of these measures could increase the risk of infection due to the emergence of new variants of the virus, as has been observed in Chile and other countries during 2022.

The results of this study suggest the need to generate new incentives for the use of personal protection measures, considering their effectiveness in preventing Covid-19 infection. Efforts could concentrate on the younger population, where adherence to Covid-19 protective measures has declined more sharply. An adequate risk communication must be performed in order to explain the importance of maintaining these preventive measures.

In addition, some incentives that can be promoted include better access to quality face masks and hand hygiene items and ensuring that capacity limits in enclosed places or on public transport are respected.

## Supporting information

**S1 Dataset.**
(DTA)

**S1 File.**
(DOCX)

## Author Contributions

**Conceptualization:** Felipe Elorrieta, Claudio Vargas, Pablo Villalobos Dintrans, Claudio Castillo, Matilde Maddaleno.

**Data curation:** Simón Varas, Yerko Martinez.

**Formal analysis:** Simón Varas, Felipe Elorrieta.

**Investigation:** Simón Varas.

**Methodology:** Claudio Vargas.

**Project administration:** Matilde Maddaleno.

**Software:** Felipe Elorrieta.

**Supervision:** Felipe Elorrieta, Claudio Vargas.

**Visualization:** Andrés Ayala.

**Writing – original draft:** Simón Varas, Felipe Elorrieta.

**Writing – review & editing:** Claudio Vargas, Pablo Villalobos Dintrans, Claudio Castillo, Yerko Martinez, Andrés Ayala, Matilde Maddaleno.

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
