## [Decision Letter · Decision Letter 0]

7 Mar 2022

PONE-D-22-04553Factors associated with change in adherence to COVID-19 personal protection measures in the Metropolitan Region, Chile.PLOS ONE

Dear Dr. Elorrieta,

Thank you for submitting your manuscript to PLOS ONE. After careful consideration, we feel that it has merit but does not fully meet PLOS ONE’s publication criteria as it currently stands. Therefore, we invite you to submit a revised version of the manuscript that addresses the points raised during the review process.

We look forward to receiving your revised manuscript.

Kind regards,

Shinya Tsuzuki, MD, MSc

Academic Editor

PLOS ONE

Journal Requirements:

Additional Editor Comments:

Both reviewers raised reasonable concerns for the manuscript and basically I agree with them. Please respond each comment appropriately before further consideration for publication.

Reviewers' comments:

Reviewer's Responses to Questions

**Comments to the Author**

1. Is the manuscript technically sound, and do the data support the conclusions?

Reviewer #1: Partly

Reviewer #2: Yes

2. Has the statistical analysis been performed appropriately and rigorously? 

Reviewer #1: I Don't Know

Reviewer #2: Yes

3. Have the authors made all data underlying the findings in their manuscript fully available?

Reviewer #1: No

Reviewer #2: Yes

4. Is the manuscript presented in an intelligible fashion and written in standard English?

Reviewer #1: Yes

Reviewer #2: Yes

5. Review Comments to the Author

Reviewer #1: The work by Varas et al. evaluates the factors associated with relaxation to personal protection measures in Chile. The work was well contextualized and with sufficient methods to achieve the proposed objectives. However, there are issues that need to be addressed before the paper can be considered for publication.

1. A thorough review of the English is required. Some sentences need to be rephrased such as “In this context, it is interesting to assess whether the self-care preventive actions have decreased their adherence in Chile.” This and some other sentences don’t make sense.

2. Reevaluate the title – “in the metropolitan area region, Chile”. The name of the city is missing.

3. Citation to Mackay (2020) - please elaborate on this as it does not seem to make sense. The page cannot be found.

4. Material and methods: please include the questionnaire in the suppl material for those interested in the research protocol.

5. Since the research was performed with human subjects, it is necessary to include information on ethics approval by an ethics committee.

6. Figure legend is lost in the text. There seems to be a problem with the text format overall.

7. Figure 2 only shows age but above it, it says age and sex. It is not clear what the probability means.

8. Figure 3 – what is predictive margin? All figure legends should be self-explanatory.

9. Text brings some discrepancies that need to be clarified. In some parts authors state that older people have lower adherence to preventive self-care measures, but in other parts, including the summary, it says that young people are more likely to relax these measures. The same is true for comparisons between men and women. In several parts of the text, it is said that women are more likely to relax physical distancing, but in the discussion authors state that “This result has also been found in other studies, showing that women are more likely to adhere to physical distancing than men”.

I also encourage authors to take a look at a similar research by Finger et al. 2021 -

https://doi.org/10.1016/j.tifs.2021.03.016

10. In the final paragraph of the Results (“For the physical distancing model, we calculated…”), the same sentence is written twice: “As presented in Figure 3, the predicted probabilities of relaxing physical distancing are consistently higher for women than for men. The likelihood of relaxing physical distance is consistently higher for women than for men.”

11. I missed a deeper discussion, comparing the results with other works, especially considering that this subject has been extensively researched.

Reviewer #2: General comments

This a very well researched study on an important topic: the prevention and control of COVID-19 in Chile. Despite the availability of potent vaccines and of late, treatment options, the use of public health preventive measures remain very critical for flattening the curve. In general, the manuscript is succinct and very well written. Nevertheless, I believe that there is some room for the authors to further improve the quality of the document which if accepted for publication could boost readership and article metrics.

I have observed an important methodological flaw of the study which would require further elaboration. Because it was conducted online, there is the possibility that the findings may only represent adherence of people who use the internet, who are more likely to be better educated and well informed. This raises a few questions which require further explanation in the manuscript. In view of this limitation, are the findings generalizable to the general population of Metropolitan Region of Chile? Could this have caused a bias in the findings and if yes, how did the authors minimized such bias? I believe that there are opportunities to correct this limitation in the revised version of the document. Are there available data on the adherence of non-internet/social media users in Chile or similar settings? Are there other studies that looked at adherence in less educated and low socioeconomic class to which your findings could be compared? Below are a few specific comments on how to improve the manuscript:

Abstract

This section is well written and could be understood as a stand-alone document. I would suggest inclusion of a sentence on your conclusion and recommendation at the end of the section.

Introduction

While this section has most of the required elements, 2-3 sentences on the COVID-19 situation and trends in the study areas should be included in the first paragraph of the section to set the tone for the study and contextualize your findings.

Methods

While this section is well written, further information on the study methods would improve the understanding and reproducibility of the study design. Furthermore, to demonstrate the validity of your findings, it is important to briefly describe how your sample size was calculated (detailed formulas and description could be included as an annex). I would suggest reorganization of this section into the following sub-sections:

Study design and setting: what type of study is this? What period was covered? The social, demographic, geographic and public health context of the study area as they relate to the subject matter? A brief introduction of the Metropolitan Health Services would also be useful to better contextualize your findings

Sampling method and sample size: how was the sample size calculated? What sampling method did you use? What was rationale for selecting the Metropolitan Region? Was it for convenience?

Data collection: description of the data collection tool (questionnaire); how many sections does it have, how many questions? This is already well presented in the current “data” sub-section.

Data analysis: This is already well presented in the current “logistic regression” sub-section.

Results

The study findings are well presented

Discussion

The discussion is well written. However, more in-depth description and rationalization of your findings would be helpful. While the authors have compared their findings to those of other studies, it would be useful to describe/discuss the factors that could have been responsible for these findings/trends in this setting. I would therefore suggest that you reorganize this section as follow for better flow and clarity:

o Paragraph 1: a very brief statement of the main objective and key findings of this study.

o Paragraphs 2-4: exhaustive discussion and rationalization of the key findings of the study. Which factors could have been responsible for the observed trends i.e. 1) the significant relaxation of the preventive measures, 2) the effect of age on the adherence to the measures, 3) the effect of gender (sex) on physical distancing and 4) the effect of vaccination? What were the findings of other similar studies? Are they comparable to your findings?

o Paragraph 5: study limitations. The authors should include the skewness of the study findings to only internet/social media as a limitation and what was done to address this or whether their findings are generalizable.

Conclusion

A conclusion section should be introduced to further elaborate on the “new incentives” which was proposed in the last paragraph of the discussion section.

6. PLOS authors have the option to publish the peer review history of their article (what does this mean?). If published, this will include your full peer review and any attached files.

Reviewer #1: No

Reviewer #2: **Yes: **Olushayo Oluseun Olu

---

## [Author Response · Author response to Decision Letter 0]

25 Mar 2022

We would like to thank the comments from the Editor and the reviewers, which have helped to substantially improve the manuscript. In what follows, we respond to each issue separately.

Reviewer #1: The work by Varas et al. evaluates the factors associated with relaxation to personal protection measures in Chile. The work was well contextualized and with sufficient methods to achieve the proposed objectives. However, there are issues that need to be addressed before the paper can be considered for publication.

1. A thorough review of the English is required. Some sentences need to be rephrased such as “In this context, it is interesting to assess whether the self-care preventive actions have decreased their adherence in Chile.” This and some other sentences don’t make sense.

We have revised the writing of the article and made some changes, which are highlighted in the document "Revised manuscript with track changes".

2. Reevaluate the title – “in the metropolitan area region, Chile”. The name of the city is missing. 

The country is administratively divided into regions. The study was conducted in the Metropolitan Region of Chile. This region is composed of 52 municipalities belonging to 6 provinces, including the city of Santiago (national capital). In the survey we obtained responses from all the provinces, so we believe it would not be correct to add in the title the name of only one city. The reasons for studying this region were outlined in the Study design and setting paragraph of the Methods and Materials section. 

3. Citation to Mackay (2020) - please elaborate on this as it does not seem to make sense. The page cannot be found. 

The link to Mackay (2020) has been corrected. In addition, the sentence that included this citation was rewritten.

4. Material and methods: please include the questionnaire in the supply material for those interested in the research protocol.

The questionnaire was included in the new section “supplementary material”

5. Since the research was performed with human subjects, it is necessary to include information on ethics approval by an ethics committee.

We include in the section “Methods and materials” a subsection called “Ethical consideration”. Here we include the approval of this project by the Ethical Review Committee of the Universidad de Santiago, Chile.

6. Figure legend is lost in the text. There seems to be a problem with the text format overall.

The caption of figures 2 and 3 was rewritten in order to make them easier to understand. 

7. Figure 2 only shows age but above it, it says age and sex. It is not clear what the probability means.

8. Figure 3 – what is predictive margin? All figure legends should be self-explanatory.

Throughout the manuscript we have replaced the term "predict margins" with "predict probabilities". Figure titles were rewritten.

9. Text brings some discrepancies that need to be clarified. In some parts authors state that older people have lower adherence to preventive self-care measures, but in other parts, including the summary, it says that young people are more likely to relax these measures. The same is true for comparisons between men and women. In several parts of the text, it is said that women are more likely to relax physical distancing, but in the discussion authors state that “This result has also been found in other studies, showing that women are more likely to adhere to physical distancing than men”.

The sentence “older people have lower adherence to preventive self-care measures” have been rewritten to “ younger people are more likely to relax adherence to preventive self-care measures”. However, it is important to note that the concepts of adherence and relaxation are not equivalent. For example, women adhere more to preventive self-care measures in 2020, but they also relaxed more and achieved similar adherence to men in 2021. We add a sentence in the same paragraph to explain this.

I also encourage authors to take a look at a similar research by Finger et al. 2021 - https://doi.org/10.1016/j.tifs.2021.03.016

This text was also incorporated to enrich the introduction of the study.

10. In the final paragraph of the Results (“For the physical distancing model, we calculated…”), the same sentence is written twice: “As presented in Figure 3, the predicted probabilities of relaxing physical distancing are consistently higher for women than for men. The likelihood of relaxing physical distance is consistently higher for women than for men.”

The second sentence of this paragraph was removed from the manuscript.

11. I missed a deeper discussion, comparing the results with other works, especially considering that this subject has been extensively researched.

We improved the discussion section. For this purpose, we added further rationalization of our findings. In addition, the discussion was enriched with a greater number of references.

Reviewer #2: General comments

This a very well researched study on an important topic: the prevention and control of COVID-19 in Chile. Despite the availability of potent vaccines and of late, treatment options, the use of public health preventive measures remain very critical for flattening the curve. In general, the manuscript is succinct and very well written. Nevertheless, I believe that there is some room for the authors to further improve the quality of the document which if accepted for publication could boost readership and article metrics.

I have observed an important methodological flaw of the study which would require further elaboration. Because it was conducted online, there is the possibility that the findings may only represent adherence of people who use the internet, who are more likely to be better educated and well informed. This raises a few questions which require further explanation in the manuscript. In view of this limitation, are the findings generalizable to the general population of Metropolitan Region of Chile? Could this have caused a bias in the findings and if yes, how did the authors minimized such bias? I believe that there are opportunities to correct this limitation in the revised version of the document. Are there available data on the adherence of non-internet/social media users in Chile or similar settings? Are there other studies that looked at adherence in less educated and low socioeconomic class to which your findings could be compared? Below are a few specific comments on how to improve the manuscript:

Abstract

This section is well written and could be understood as a stand-alone document. I would suggest inclusion of a sentence on your conclusion and recommendation at the end of the section.

We included a sentence in the abstract about our conclusions and recommendations.

Introduction

While this section has most of the required elements, 2-3 sentences on the COVID-19 situation and trends in the study areas should be included in the first paragraph of the section to set the tone for the study and contextualize your findings.

In the Introduction section, we included a paragraph mentioning the context of the virus circulation and confinement conditions in the Metropolitan Region at the moment the survey was carried out.

Methods

While this section is well written, further information on the study methods would improve the understanding and reproducibility of the study design. Furthermore, to demonstrate the validity of your findings, it is important to briefly describe how your sample size was calculated (detailed formulas and description could be included as an annex). I would suggest reorganization of this section into the following sub-sections:

Study design and setting: what type of study is this? What period was covered? The social, demographic, geographic and public health context of the study area as they relate to the subject matter? A brief introduction of the Metropolitan Health Services would also be useful to better contextualize your findings

Sampling method and sample size: how was the sample size calculated? What sampling method did you use? What was rationale for selecting the Metropolitan Region? Was it for convenience?

Data collection: description of the data collection tool (questionnaire); how many sections does it have, how many questions? This is already well presented in the current “data” sub-section.

Data analysis: This is already well presented in the current “logistic regression” sub-section.

The methods section has been restructured according to these recommendations. Consequently, we add four sub-sections: Study design and setting, sampling method and sample size, data collection and data analysis.

Results

The study findings are well presented

Discussion

The discussion is well written. However, more in-depth description and rationalization of your findings would be helpful. While the authors have compared their findings to those of other studies, it would be useful to describe/discuss the factors that could have been responsible for these findings/trends in this setting. I would therefore suggest that you reorganize this section as follow for better flow and clarity:

o Paragraph 1: a very brief statement of the main objective and key findings of this study.

o Paragraphs 2-4: exhaustive discussion and rationalization of the key findings of the study. Which factors could have been responsible for the observed trends i.e. 1) the significant relaxation of the preventive measures, 2) the effect of age on the adherence to the measures, 3) the effect of gender (sex) on physical distancing and 4) the effect of vaccination? What were the findings of other similar studies? Are they comparable to your findings?

o Paragraph 5: study limitations. The authors should include the skewness of the study findings to only internet/social media as a limitation and what was done to address this or whether their findings are generalizable.

According to this comment, the discussion section was reorganized as follows:

Paragraph 1: Main objective and the key finding of this study.

Paragraph 2: The effect of age on the adherence to the three self-care preventive measures.

Paragraph 3: The effect of sex and vaccination on relaxation the adherence of physical distance.

Paragraph 4: Limitations of this study.

Paragraph 5: Limitation of the survey due to the collection of responses through social networks. 

In addition, we add a more detailed discussion of each of our findings, with further references to other similar studies.

Conclusion

A conclusion section should be introduced to further elaborate on the “new incentives” which was proposed in the last paragraph of the discussion section.

Section 5 "Conclusions" has been included in the manuscript. In this section, we include some examples of incentives that can be promoted in order to maintain the adherence to these measures.

---

## [Decision Letter · Decision Letter 1]

8 Apr 2022

Factors associated with change in adherence to COVID-19 personal protection measures in the Metropolitan Region, Chile.

PONE-D-22-04553R1

Dear Dr. Elorrieta,

We’re pleased to inform you that your manuscript has been judged scientifically suitable for publication and will be formally accepted for publication once it meets all outstanding technical requirements.

Kind regards,

Shinya Tsuzuki, MD, MSc

Academic Editor

PLOS ONE

Additional Editor Comments (optional):

Reviewers' comments:

Reviewer's Responses to Questions

**Comments to the Author**

1. If the authors have adequately addressed your comments raised in a previous round of review and you feel that this manuscript is now acceptable for publication, you may indicate that here to bypass the “Comments to the Author” section, enter your conflict of interest statement in the “Confidential to Editor” section, and submit your "Accept" recommendation.

Reviewer #1: All comments have been addressed

Reviewer #2: All comments have been addressed

2. Is the manuscript technically sound, and do the data support the conclusions?

Reviewer #1: (No Response)

Reviewer #2: Yes

3. Has the statistical analysis been performed appropriately and rigorously? 

Reviewer #1: (No Response)

Reviewer #2: Yes

4. Have the authors made all data underlying the findings in their manuscript fully available?

Reviewer #1: (No Response)

Reviewer #2: Yes

5. Is the manuscript presented in an intelligible fashion and written in standard English?

Reviewer #1: (No Response)

Reviewer #2: Yes

6. Review Comments to the Author

Reviewer #1: Authors have addressed my concerns and updated the manuscript accordingly. Paper can be accepted for publication.

Reviewer #2: (No Response)

7. PLOS authors have the option to publish the peer review history of their article (what does this mean?). If published, this will include your full peer review and any attached files.

Reviewer #1: No

Reviewer #2: **Yes: **Olushayo Olu

---

## [Editor Report · Acceptance letter]

4 May 2022

PONE-D-22-04553R1 

Factors associated with change in adherence to COVID-19 personal protection measures in the Metropolitan Region, Chile. 

Dear Dr. Elorrieta:

I'm pleased to inform you that your manuscript has been deemed suitable for publication in PLOS ONE. Congratulations! Your manuscript is now with our production department. 

Kind regards, 

on behalf of

Dr. Shinya Tsuzuki 

Academic Editor

PLOS ONE